# Etiology of Carpal Tunnel Syndrome in a Large Cohort of Children

**DOI:** 10.3390/children8080624

**Published:** 2021-07-23

**Authors:** Christina T. Rüsch, Ursula Knirsch, Daniel M. Weber, Marianne Rohrbach, André Eichenberger, Jürg Lütschg, Kirsten Weber, Philip J. Broser, Georg M. Stettner

**Affiliations:** 1Neuromuscular Center Zurich and Department of Pediatric Neurology, University Children’s Hospital Zurich, University of Zurich, 8032 Zurich, Switzerland; Christina.Ruesch@kispi.uzh.ch (C.T.R.); Ursula.Knirsch@kispi.uzh.ch (U.K.); 2Division of Pediatric Neurology, Children’s Hospital of Eastern Switzerland, 9006 St. Gallen, Switzerland; juerg.luetschg@unibas.ch (J.L.); PhilipJulian.Broser@kispisg.ch (P.J.B.); 3Division of Hand Surgery, University Children’s Hospital Zurich, University of Zurich, 8032 Zurich, Switzerland; Daniel.Weber@kispi.uzh.ch; 4Division of Metabolism, University Children’s Hospital Zurich, University of Zurich, 8032 Zurich, Switzerland; Marianne.Rohrbach@kispi.uzh.ch; 5Division of Radiology, University Children’s Hospital Zurich, University of Zurich, 8032 Zurich, Switzerland; Andre.Eichenberger@kispi.uzh.ch; 6Division of Hand Surgery, Children’s Hospital of Eastern Switzerland, 9006 St. Gallen, Switzerland; kirsten.weber@kispisg.ch

**Keywords:** carpal tunnel syndrome, median nerve neuropathy, electrodiagnostic studies, neuromuscular ultrasound, mucopolysaccharidosis

## Abstract

(1) Background: Carpal tunnel syndrome (CTS), a compressive mononeuropathy of the median nerve at the wrist, is rare in childhood and occurs most frequently due to secondary causes. (2) Methods: Medical history, electrodiagnostic findings, and imaging data of patients with CTS from two pediatric neuromuscular centers were analyzed retrospectively. The etiology of CTS was investigated and compared with the literature. (3) Results: We report on a cohort of 38 CTS patients (*n* = 22 females, *n* = 29 bilateral, mean age at diagnosis 9.8 years). Electrodiagnostic studies of all patients revealed slowing of the antidromic sensory or orthodromic mixed nerve conduction velocities across the carpal tunnel or lack of the sensory nerve action potential and/or prolonged distal motor latencies. Median nerve ultrasound was diagnostic for CTS and confirmed tumorous and vascular malformations. Etiology was secondary in most patients (*n* = 29; 76%), and mucopolysaccharidosis was the most frequent underlying condition (*n* = 14; 37%). Idiopathic CTS was rare in this pediatric cohort (*n* = 9; 24%). (4) Conclusion: Since CTS in childhood is predominantly caused by an underlying disorder, a thorough evaluation and search for a causative condition is recommended in this age group.

## 1. Introduction

Carpal tunnel syndrome (CTS) is a compressive mononeuropathy of the median nerve at the wrist. In contrast to CTS in adult patients, the condition in childhood is rare, often manifests with atypical symptoms, and most frequently occurs secondarily due to other causes. In children, CTS was first described by Martin and Mass in 1958 [1], who reported on three children with recurrent episodes of hand pain. In 1989 Poilvach [2] carried out an extensive literature search and presented 52 cases of childhood CTS. He suggested the first etiopathological classification of the various underlying causes. Van Meir and De Smet [3,4] continued this work and performed a meta-analysis of 163 cases from 35 articles, mostly case reports or small case series.

The diagnosis of CTS in adults is primarily based on clinical symptoms and can be confirmed with electrodiagnostic studies [5]. In children, symptoms are often atypical, which reinforces the importance of technical investigations. Regardless of the etiology, isolated slowing of sensory or mixed nerve conduction velocity and/or prolongation of the distal motor latency (DML) of the median nerve across the carpal tunnel are electrophysiological hallmarks for CTS. Recently, neuromuscular ultrasound has been recognized as a valuable method for different neuromuscular conditions including entrapment neuropathies. This applies also to the evaluation of CTS [6]. Characteristics of median nerve ultrasound studies consist of an increase of both the cross-sectional area (CSA) at the wrist and the wrist-to-forearm ratio (WFR) [6]. For the majority of pediatric CTS cases, an underlying cause can be found, in particular hereditary metabolic conditions with mucopolysaccharidoses and mucolipidoses as the largest disease group, followed by congenital malformations, connectivopathies, endocrinopathies, and acquired lesions like malignancies or tumor-like and traumatic lesions [2,3,7,8,9].

The aim of this study was to investigate the etiology of CTS in a cohort from two tertiary pediatric neuromuscular centers in Switzerland (University Children’s Hospital Zurich and Children’s Hospital of Eastern Switzerland St. Gallen, Switzerland). We retrospectively analyzed the data of pediatric patients with CTS and evaluated diagnostic procedures and findings.

## 2. Materials and Methods

We retrospectively analyzed data of patients diagnosed with CTS in two tertiary pediatric neuromuscular centers in Switzerland (University Children’s Hospital Zurich and Children’s Hospital of Eastern Switzerland St. Gallen). Patients with an age below 18 years at diagnosis of CTS with characteristic electrophysiological findings were included in our study. The main focus of this study was to investigate the etiology of childhood CTS. Therefore, demographic data, medical history, manifesting symptoms, examination findings, underlying conditions, and proportion of etiologies were analyzed. For identification of CTS patients, the clinical information system of the two participating centers and registers of electrophysiological and surgical interventions were screened for the diagnosis of CTS. All patients who were diagnosed with CTS in the years 2005–2020 at the University Children’s Hospital Zurich and 2016–2020 at the Children’s Hospital of Eastern Switzerland, St. Gallen were included. All patients gave their consent to be included in our study.

For inclusion, all patients had to fulfill standard electrodiagnostic criteria for CTS. Since this is a retrospective work, different electrophysiological standard procedures established for the investigation of adults were performed [10]. Midpalm stimulation of the median and ulnar nerves and determination of latency differences between the orthodromic mixed nerve potentials at the wrist at a distance of 6–8 cm was preferred, because this method is least dependent on the small size of the hand in younger children, in which standard distal distances used in other electrodiagnostic approaches sometimes cannot be respected. Any latency difference, referred to as “palmdiff”, of ≥0.4 ms was considered diagnostic [10]. Alternatively, fractioned antidromic sensory nerve conduction studies with stimulation at the wrist and midpalm and recording of sensory nerve action potentials on the second digit were performed. Slowing of the sensory nerve conduction velocity of ≥10 m/s across the carpal tunnel was considered diagnostic for CTS. In addition, distal motor latencies (DML) of the compound muscle action potential (CMAP) recorded from the abductor pollicis brevis muscle were obtained after median nerve stimulation at the wrist with a distance of 7.0 cm whenever possible. A DML of ≥4.1 ms was considered diagnostic [10]. Two different ENMG systems were used in our centers: Viking Monograph (Nicolet Biomedical Inc. Madison, WI, USA; used in Zurich) and System Plus (Micromed, Venice, Italy; used in St. Gallen).

In addition, ultrasound imaging data of the median nerve were analyzed if available.

Median nerve ultrasound (US) imaging was done in 24% of the patients by a pediatric radiologist and/or pediatric neurologist trained in peripheral nerve US following standard procedures [6]. The presence of structural changes was investigated along the median nerve. In addition, the median nerve cross-sectional area (CSA) at standard locations was measured and the wrist-to-forearm ratio (WFR) calculated with reference to age related nerve US normal values [11,12]. In both centers a Canon Aplio i800 (Canon Medical Systems, Tokyo, Japan) ultrasound imaging system, equipped with i33LX9, i24LX8, i18LX5, and i22LH8 was used.

The study was approved by the local ethics committee on 23 June 2020, and registered with the Swiss project database (BASEC 2020-01016). Written informed consent was obtained from the caregivers prior to inclusion of participants in the study.

## 3. Results

### 3.1. Demographics

We identified 38 patients (*n* = 22 females, *n* = 16 males) diagnosed with CTS in the two pediatric neuromuscular centers between 2005 and 2020 in Zurich and 2016 and 2020 in St. Gallen. The demographics of the patients are shown in Table 1. See Appendix A for more detailed individual information.

### 3.2. Etiology

Lysosomal storage diseases (mucopolysaccharidosis and mucolipidosis) were the most frequent underlying conditions in our cohort (*n* = 15; 39%). Five patients (13%) showed CTS associated with a hereditary neuropathy (*n* = 3 probable hereditary neuropathy with liability to pressure palsy (HNPP) with positive family history, *n* = 1 Charcot-Marie-Tooth CMT type 1A, *n* = 1 associated with autosomal recessive spastic ataxia of Charlevoix-Saguenay (ARSACS)). In the three patients with a positive family history for HNPP the parents did not consent to genetic testing for their children. We identified CTS due to congenital malformations in four patients (11%). Two of them had been diagnosed with geleophysic dysplasia, one with the ultra-rare condition melorheostosis and one with a hemihypertrophia syndrome of unknown etiology. In two patients (5%), CTS occurred due to a benign tumor (*n* = 2 perineurioma, Figure 1), of which one perineurioma was associated with a PIK3CA gene mutation. A posttraumatic CTS was found in two patients (5%). One patient (3%) suffered from bilateral CTS associated with rheumatoid arthritis. Altogether, a secondary CTS etiology was confirmed in 29 patients (76%). In only nine patients (24%) was CTS considered idiopathic because of the absence of other explaining findings. See Table 2 for a summary of the CTS etiology.

### 3.3. Clinical Findings

Most patients indicated typical complaints of a CTS, e.g., paraesthesia and/or dysaesthesia. However, only 29% (4/14) of CTS patients with MPS indicated complaints related to the CTS, although thenar muscle atrophy was already present at diagnosis in 86% (12/14) of these patients. Thenar muscle atrophy was also present at the time of diagnosis of CTS at a high proportion in most of the other conditions: congenital malformations 3/4, tumors 2/2, traumatic lesion 1/2, rheumatoid arthritis 1/1. Only in CTS associated with neuropathy was thenar muscle atrophy not observed (0/5), and idiopathic CTS showed thenar muscle atrophy only in 3/9 patients. See Appendix A for more detailed individual information.

### 3.4. Electrophysiological Examination

Electrodiagnostic studies of all patients revealed a significant latency difference between orthodromic median and ulnar mixed nerve potentials and/or slowing of the antidromic median sensory nerve conduction velocities across the carpal tunnel or lack of the sensory nerve action potential and/or prolonged median DML. See Appendix A for detailed information.

### 3.5. Ultrasound Imaging

In nine patients US imaging was performed. In all CTS patients, the WFR ratio and/or the CSA of the median nerve at the wrist was increased. In addition, structural lesions of the median nerve were reliably detected. See Appendix A for details. In the two patients with perineurioma, the echogenicity and structure of the nerve was altered (enlarged fascicles, increased perineuronal tissue). In these patients an MRI of the wrist and forearm was also performed and confirmed the US findings. The final diagnosis was then confirmed by histological examination following incisional biopsy during surgical decompression of the carpal tunnel.

## 4. Discussion

Compared to CTS in adults, CTS in children is rare. However, since children may not present with typical symptoms and may, in part, not communicate their complaints depending on their developmental stage and/or cognitive impairment, CTS is possibly underdiagnosed in this age group. Nevertheless, it is important to consider the presence of CTS even in toddlers with atypical symptoms, because the majority of CTS is caused by an underlying condition and requires early surgical treatment in order to prevent axonal median nerve damage.

In our cohort the age range at diagnosis was 2.5 to 17 years. The youngest child reported with CTS was 9 months old [13]. CTS was bilateral in 76% of our cohort, and a bilateral manifestation occurred mostly in CTS with an underlying hereditary disorder. In comparison, bilateral CTS at manifestation occurs only in approximately 50% of the adult population [14]. A surgical intervention was performed in 71% of our cohort. This high rate of surgical interventions was also related to the secondary nature of childhood CTS. Almost all children from our cohort who harbored a hereditary condition (e.g., lysosomal storage diseases, congenital malformations) or a tumor associated with CTS required surgical intervention because of the low likelihood of improvement under conservative treatment due to the stationary or progressive nature of these conditions.

Mucopolysaccharidosis was the most common cause of CTS in our cohort. A high prevalence of mucopolysaccharidosis in childhood CTS is also reported in the literature [3,8,15]. MPS constitutes a group of rare lysosomal storage diseases with multisystem manifestation. CTS is a common musculoskeletal manifestation of MPS [9,16,17,18]. The symptoms of CTS in patients with MPS are, however, often not as distinct as in other etiologies. In our cohort, less than 30% indicated complaints related to CTS. The early nonspecific symptoms of CTS in MPS, compounded with communication barriers due to age and intellectual disability, often lead to delayed diagnosis with thenar wasting and potential permanent loss of hand function [17,18]. Therefore, routine biannual physical examination and annual electrophysiological screening for CTS is recommended in the care standards for MPS even in the absence of suggestive symptoms [17]. Adhering to this recommendation, CTS was diagnosed in MPS patients at an early stage in our cohort, and surgical intervention was performed in all MPS CTS patients. Three of 14 MPS patients showed recurrent CTS within 3–11 years after the first surgical intervention. Follow up investigations showed normalization of the nerve conduction studies only in three patients after carpal tunnel release. These three patients were identified and treated very early (below 5 years of age). The patients with a later surgical intervention showed chronic axonal damage of the median nerve. These findings confirm the importance of physical examinations every six months and annual electrodiagnostic screening, which enables early diagnosis and treatment of CTS in the MPS population. As reviewed by Patel et al. [17], MPS patients are at risk of developing CTS very early in life. In fact, the youngest patient from our cohort, diagnosed with CTS at 30 months of age, belongs to the MPS patient group. Screening for CTS, therefore, should be initiated immediately after the diagnosis of MPS and continued frequently thereafter.

Polyneuropathy was the reason for CTS in five patients (13%) in our cohort. One patient suffered from CMT1A and one patient from sensorimotor neuropathy associated with ARSACS. In one patient, HNPP was assumed as causative for CTS because of a genetically confirmed HNPP in the child’s mother. Two additional patients with early onset CTS were siblings, and family history revealed the presence of HNPP over several generations. The parents of these three patients did not consent to the genetic confirmation of HNPP in their children. Del Colle [19] describes an identical constellation compared to the family with two affected siblings in our cohort: A family with HNPP in several generations and a high prevalence of early onset CTS, in some cases as the only manifestation of the HNPP. In general, bilaterally prolonged DML of the median nerve, prolonged DML and/or reduced motor nerve conduction velocities in the peroneal nerve and sensory nerve conduction velocity slowing are highly suggestive of HNPP when there is a positive family history of polyneuropathy [20].

CTS associated with congenital malformations was the fourth most common etiology in our cohort (affecting 11%). Our cohort includes one individual with melorheostosis, an extremely rare and progressive bone disease accompanied by hyperostosis and soft tissue fibrosis. Hand involvement had only been reported sporadically in this condition [21,22,23]. Interestingly, our cohort also includes two patients with geleophysic dysplasia, a rare hereditary condition characterized by severe short stature, short extremities, progressive joint limitation, thickened skin, and pseudomuscular build. Together with acromicric dysplasia, the geleophysic dysplasia belongs to the acromelic dysplasia group. These two conditions share, in part, similarities of the genetic pathway and phenotype. Hand involvement causes an increased risk for the development of CTS, which might be as high as 35% in geleophysic and acromicric dysplasias [24].

In our cohort we found two patients with an intraneural perineurioma, a rare benign peripheral nerve sheath tumor, which has only been included in the WHO classification system since 2000 [25]. In both patients the diagnosis of the tumor was suspected in the US investigation which followed the electrophysiological diagnosis of CTS. A histopathological examination confirmed the diagnosis in these two patients. Molecular investigation performed with biopsy material showed a pathogenic somatic mutation in the PIK3CA gene in one patient. Perineuriomatous pseudo-onion bulb proliferation is considered part of the PIK3CA-related overgrowth spectrum (PROS) [26] and has also been described in lipomatosis of peripheral nerves with or without nerve territory overgrowth in association with PIK3CA mutations [27]. Dailiana et al. [28] published a case series of tumors and tumor-like lesions affecting the median nerve as rare causes for CTS. However, most of the patients in this study showed nerve compression due to extraneural masses.

In nine patients (24%), the CTS was classified as idiopathic and no obvious underlying condition could be confirmed. This etiological group included one child with a bilateral CTS and the additional diagnosis of familial Mediterranean fever (FMF), who was under colchicine treatment for 18 months prior to the manifestation of bilateral CTS. Since it is known that colchicine can cause polyneuropathies amongst other side effects, Isikay et al. [29] examined a group of 88 children with FMF under Colchicine treatment and found only one patient with CTS. In addition, only Bademci et al. [30] described a bilateral CTS in a young woman with FMF. Due to the fact of the high incidence of both FMF in some populations and CTS in general, this association might be random, which is also the conclusion of a large retrospective study of comorbidities in 2000 FMF patients including more than 600 children [31]. Our patient with bilateral CTS and FMF might nevertheless be an example for the suspicion that even the low proportion of idiopathic CTS in children might be overestimated, because this etiological group most likely includes patients with underlying disorders that are unknown or not detectable at the time of CTS manifestation.

Neuromuscular US is becoming a standard investigation in the evaluation of peripheral nerve and muscle diseases, including CTS. In addition to electrodiagnostic procedures, detection of median nerve enlargement at the wrist by US has been suggested as a sensitive and valuable diagnostic method [32,33,34]. Billakota et al. [6] performed a large retrospective analysis of median nerve ultrasound investigations in CTS and concluded that median nerve US is nearly as sensitive as electrophysiological testing, which is considered the diagnostic gold standard in CTS. Bäumer et al. [35] specifically examined the value of US in the management of patients with MPS. In their study, US had an even higher sensitivity for the detection of CTS compared to electrophysiology. In our cohort, which focused on the etiology and not on the diagnostic measures, US was performed only in a small number of patients. This is primarily a consequence of the retrospective nature of this study. Even the low number of median nerve US investigations in our cohort, however, demonstrates that the increase of both the CSA of the median nerve at the wrist and the WFR are also indicative of CTS in children. In addition, nerve US is a sensitive method to detect tumor associated median nerve lesions. Therefore, US is a valuable tool to support the clinical and electrophysiological diagnosis of CTS, especially in children since it is a quick, non-invasive and painless method.

In conclusion, we were able to identify a broad spectrum of underlying etiologies in our cohort of childhood CTS. Our study confirms that idiopathic CTS in children is rare and most commonly secondary to an underlying condition with mucopolysaccharidosis as the most common cause.

To prevent delayed diagnosis, which can lead to thenar wasting and permanent loss of hand function, we propose the following diagnostic algorithm for patients at risk and patients with symptoms suggestive of CTS:

Patients with conditions associated with a high risk for CTS should be clinically screened for symptoms of CTS at the time of the primary diagnosis and frequently thereafter. Thenar muscle atrophy, pain and/or sensory symptoms, and/or disturbances of nail growth in digits I to III, a positive Tinel sign at the wrist and/or deterioration of dexterity are features potentially pointing to the presence of CTS. Physical examination is recommended as frequent as every 6 months for MPS patients, the largest patient group at risk for CTS [17,18]. In addition to lysosomal storage diseases, several genetic conditions, including HNPP and other hereditary neuropathies and congenital malformation syndromes like acromelic dysplasia, melorheostasis, and hemihypertrophia syndromes have a high risk for early CTS, too, and should also be screened for CTS by frequent physical examination. It is, however, important to emphasize, that CTS symptoms in children are often atypical and complaints might not be communicated due to the developmental stage and/or cognitive impairment of these patients. Therefore, physical examination and screening for CTS should be supplemented by median nerve US. Our study shows that the early conduct of median nerve US might be diagnostic for childhood CTS. In addition, all patients at risk should undergo annual or even more frequent electrodiagnostic investigations, applying standard procedures for the investigation of CTS. If these investigations prove the presence of CTS, early surgical intervention should be discussed because conservative treatment might not be effective due to the stationary or progressive nature of the primary conditions and the high risk of axonal median nerve damage.

Patients with symptoms suggestive of CTS without known underlying conditions should undergo the same procedures consisting of physical examination, median nerve US, and electrodiagnostic testing. Due to the frequent secondary nature of childhood CTS, a thorough investigation and search for an underlying disease is mandatory.

## Figures and Tables

**Figure 1 children-08-00624-f001:**
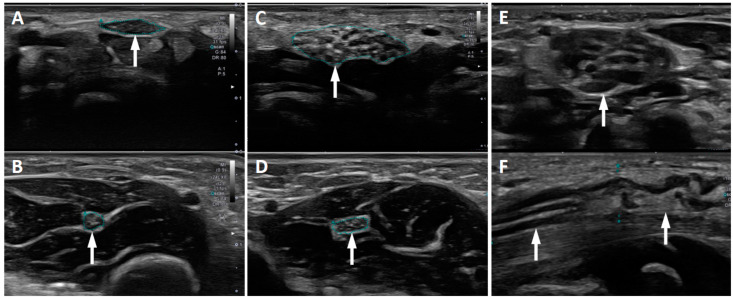
Ultrasound of the median nerve in childhood CTS. (**A**,**B**) Idiopathic CTS with transverse sonogram of the median nerve at wrist (**A**) and forearm (**B**). The median nerve ultrasound investigation demonstrated a pathologically increased WFR of 2.3. (**C**–**F**) Intraneural perineurioma in two patients: Transverse sonograms of the median nerve at wrist (**C**) and forearm (**D**) of one patient, and transverse (**E**) and longitudinal (**F**) sonogram at the wrist of the second patient with intraneural perineurioma.

**Table 1 children-08-00624-t001:** Demographics of childhood CTS patients.

Gender [*n* (%)]	female	22 (58%)
	male	16 (42%)
Age at diagnosis [years]	mean	9.8
	range	2.5–17
Location [*n* (%)]	unilateral	9 (24%)
	bilateral	29 (76%)
Positive family history for CTS [*n* (%)]		8 (21%)
Treatment [*n* (%)]	conservative	11 (29%)
	surgical	27 (71%)

**Table 2 children-08-00624-t002:** Etiology of CTS.

		Number of Patients (*n*)	Unilateral vs. Bilateral
Lysosomal storage diseases	MPS Type 1	4	bilateral
	MPS Type 2	7	bilateral
	MPS Type 3	1	bilateral
	MPS Type 6	2	bilateral
	Mucolipidosis Type 3	1	bilateral
Neuropathy	HNPP (assumed)	3	bilateral
	CMT1A	1	bilateral
	associated with ARSACS	1	bilateral
Congenital malformation	Geleophysic dysplasia	2	bilateral
	Melorheostosis	1	unilateral
	Hemihypertrophia	1	bilateral
Tumor	Intraneural Perineurioma	2	unilateral
Traumatic lesion		2	unilateral
Rheumatoid arthritis		1	bilateral
Idiopathic		9	4 unilateral, 5 bilateral

ARSACS = autosomal recessive spastic ataxia of Charlevoix-Saguenay; CMT1A = Charcot-Marie-Tooth disease type 1A; HNPP = hereditary neuropathy with liability to pressure palsy; MPS = mucopolysaccharidosis.

## Data Availability

Data is contained within the article or Appendix A.

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
