# Peer review of "Etiology of Carpal Tunnel Syndrome in a Large Cohort of Children"

_children, 2021, doi:10.3390/children8080624_

Round 1

Reviewer 1 Report

This is a retrospective study on CTS etiology, diagnosis and treatment in a cohort of 38 pediatric patients.

Mucopolysaccharidoses and mucolipidosis were found the most commonly reported etiologies for CTS, similarly to the literature. However, it should be highlighted that the early nonspecific symptoms, compounded with communication barriers due to age and intellectual disability could lead to delayed diagnosis and potential permanent loss of hand function. Most centers that treat MPS patients perform screening for the presence of CTS, including a combination of regular physical examination and neurophysiological tests such as nerve conduction studies, nerve ultrasonography, or electromyography. I propose to read the study of Patel et al, 2020;35(6):410-417.

The strenght of this study is given by the quite large cohort of patients, regarding pediatric population and rarity of this phenomenon.

A great value of this manuscript was given by presence of neuromuscular US examination as a quick, simple, and non-inasive method of CTS diagnosis, however it was performed only in 9 (24 %) patients.

I propose to replace the term ''metabolic storage diseases'' into lysosomal storage diseases.

Author Response

Response to Reviewer 1 Comments:

We thank Reviewer 1 for the valuable comments. We adapted the manuscript according to the reviewer’s comments.

Point 1: Mucopolysaccharidoses and mucolipidosis were found the most commonly reported etiologies for CTS, similarly to the literature. However, it should be highlighted that the early nonspecific symptoms, compounded with communication barriers due to age and intellectual disability could lead to delayed diagnosis and potential permanent loss of hand function. Most centers that treat MPS patients perform screening for the presence of CTS, including a combination of regular physical examination and neurophysiological tests such as nerve conduction studies, nerve ultrasonography, or electromyography. I propose to read the study of Patel et al, 2020;35(6):410-417.

Response 1: We read the study of Patel et al. again, which already was cited in the first version of the submitted manuscript. The study of Patel et al. certainly is a reference for the evaluation and care of CTS in MPS patients. We adapted the description of the standard procedures for the evaluation of CTS in MPS patients in the Discussion according to this reference (lines 209 ff.). According to the reviewer’s comment we also highlighted the early nonspecific symptoms of CTS in MPS in the Discussion (lines 206-209) in addition to the comment, that less than 30% of MPS patients in our cohort indicated complaints related to CTS at the time of CTS diagnosis (lines 205-206, unchanged from the first version of the submitted manuscript). Prompted by a comment of Reviewer 2 we now propose  a diagnostic algorithm for CTS in children at the end of the Discussion, in which we also refer to the work of Patel et al.

Point 2: A great value of this manuscript was given by presence of neuromuscular US examination as a quick, simple, and non-invasive method of CTS diagnosis, however it was performed only in 9 (24 %) patients.

Response 2: We thank the reviewer for pointing out the great value of the neuromuscular US in the diagnostic procedures for CTS. The low number of US investigations in our cohort is a limitation of our retrospective study (discussed in lines 301 ff). Even the low number of median nerve US investigations in our cohort, however, demonstrates that US is a valuable method to support the clinical and electrophysiological diagnosis of CTS especially in children. Within our institutions, neuromuscular US has become standard for the evaluation of CTS.

Point 3: I propose to replace the term ''metabolic storage diseases'' into lysosomal storage diseases.

Response 3: In the revised manuscript we now use “lysosomal storage disease”.

Reviewer 2 Report

The authors describe a series of children with CTS. Most were secondary.

The authors should clarify what this series adds to the literature or how it affects our current practice. Though this is indeed rare and usually secondary in the pediatric population, this has been extensively described.

Maybe a diagnostic algorithm?  Try to describe the patients most at risk for developing early CTS?

Author Response

We thank Reviewer 2 for the valuable comments. We adapted the manuscript according to the reviewer’s comments.

Point 1: The authors should clarify what this series adds to the literature or how it affects our current practice. Though this is indeed rare and usually secondary in the pediatric population, this has been extensively described.

Response 1: We retrospectively investigated the etiology of CTS patients in two Swiss pediatric neuromuscular center. This study confirms previous reports and demonstrates, that CTS in childhood is secondary in the majority, with MPS as the largest disease group. CTS in children may not present with typical symptoms. In particular in MPS patients, less than 30% of MPS patients in our cohort indicated complaints related to CTS at the time of CTS diagnosis (refer to lines 205 ff.). Because CTS may lead to a permanent deterioration of the motor function of the hand, we consider it important to raise the awareness a) for CTS in children in general, b) for the increased risk of early CTS in certain patient groups and c) for the fact, that CTS in children may occur with minor or atypical symptoms. We agree, that the results of our study are rather of confirmatory nature.

Although neuromuscular ultrasound has become a standard method in adult neuromuscular care including CTS, only few reports exist describing its use in CTS in the pediatric population. Only recently normal values for nerve US in children have been published (s. References 11, 12, 35). Although performed only in a fraction of patients in our cohort, median nerve ultrasound proofed to be a valuable method to support the clinical and electrophysiological diagnosis of CTS especially in children (please refer to the Discussion lines 293 ff). The observation of our study should alter the current practice, and median nerve US should be included in the investigation of pediatric patients at risk for CTS or with symptoms suggestive for CTS as a standard procedure. In the diagnostic algorithm, which we added to the Discussion (s. Point 2 below), we suggest to implement median nerve US early in the investigatory process of CTS (lines 330 ff).

Point 2: Maybe a diagnostic algorithm?  Try to describe the patients most at risk for developing early CTS?

Response 2: According to the reviewer’s suggestion we expanded the Discussion and propose a diagnostic algorithm for patients at risk for CTS and patients with symptoms suggestive for CTS (lines 318-350). Within this diagnostic algorithm we refer to the largest patient group at risk for developing early CTS (MPS patients). In addition, we list other important conditions with an increased risk for an early CTS, e.g. neuropathies and congenital malformation syndromes. Within the Discussion about CTS in MPS, we now emphasize, that patients with MPS are at risk for developing CTS very early in life (lines 231-234).